# Sex-Differential Impact of Human Cytomegalovirus Infection on In Vitro Reactivity to Toll-Like Receptor 2, 4 and 7/8 Stimulation in Gambian Infants

**DOI:** 10.3390/vaccines8030407

**Published:** 2020-07-22

**Authors:** Momodou Cox, Jane U. Adetifa, Fatou Noho-Konteh, Lady C. Sanyang, Abdoulie Drammeh, Magdalena Plebanski, Hilton C. Whittle, Sarah L. Rowland-Jones, Iain Robertson, Katie L. Flanagan

**Affiliations:** 1Infant Immunology Group, Vaccines and Immunity Theme, MRC Unit, P.O. Box 273, Fajara, The Gambia; momodou.cox@student.rmit.edu.au (M.C.); jane_osa@yahoo.com (J.U.A.); fatounohokonteh@gmail.com (F.N.-K.); csanyang@mrc.gm (L.C.S.); abdrammeh@mrc.gm (A.D.); hcwhittle@yahoo.co.uk (H.C.W.); sarah.rowland-jones@ndm.ox.ac.uk (S.L.R.-J.); 2School of Health & Biomedical Science, RMIT University, Melbourne, VIC 3083, Australia; magdalena.plebanski@rmit.edu.au; 3Department of Immunology and Pathology, Monash University, Melbourne, VIC 3004, Australia; 4Faculty of Infectious and Tropical Diseases, London School of Hygiene and Tropical Medicine, London WC1E 7HT, UK; 5Nuffield Department of Medicine, University of Oxford, Oxford OX3 9DU, UK; 6School of Medicine and School of Health Sciences, University of Tasmania, Launceston, Tasmania 7250, Australia; iain.robertson@utas.edu.au

**Keywords:** human cytomegalovirus, human infants, innate immunity, sex differences, vaccination, toll-like receptor, adjuvants, cytokines

## Abstract

Human cytomegalovirus (HCMV) infection rates approach 100% by the first year of life in low-income countries. It is not known if this drives changes to innate immunity in early life and thereby altered immune reactivity to infections and vaccines. Given the panoply of sex differences in immunity, it is feasible that any immunological effects of HCMV would differ in males and females. We analysed ex vivo innate cytokine responses to a panel of toll-like receptor (TLR) ligands in 108 nine-month-old Gambian males and females participating in a vaccine trial. We found evidence that HCMV suppressed reactivity to TLR2 and TLR7/8 stimulation in females but not males. This is likely to contribute to sex differences in responses to infections and vaccines in early life and has implications for the development of TLR ligands as vaccine adjuvants. Development of an effective HCMV vaccine would be able to circumvent some of these potentially negative effects of HCMV infection in childhood.

## 1. Introduction

The human herpes virus 5, more commonly known as human cytomegalovirus (HCMV), is known to infect a high proportion of children in low-income settings in the first year of life [1]. Acutely infected individuals go on to develop a chronic latent infection that is generally asymptomatic, although it can reactivate throughout life, particularly during times of immune suppression [2]. Congenital HCMV infection leads to a series of neurodevelopmental defects including deafness and cognitive impairment [3]. While infection after the neonatal period is not thought to lead to these serious outcomes, the clinical and immunological implications of establishing HCMV infection in early life remain uncertain.

Infection with HCMV activates the production of antiviral type I interferons (IFNs) by dendritic cells (DCs) [4]. While TLR2 recognizes HCMV, TLR2 does not appear to be responsible for triggering an antiviral type 1 IFN response [5,6]. Indeed, MCMV studies suggest that toll-like receptors (TLR) 3, 7 and 9 act redundantly to promote the early initial type I interferon, although TLR9 probably plays a greater role and TLR2 signalling may also be involved [7]. The dendritic cell (DC)-derived type I IFNs activate NF-κB via intracellular signalling pathways, leading to the expression of multiple IFN-responsive genes with antiviral properties and the production of innate pro-inflammatory cytokines including TNF-α, IL-1β and IL-6 [8]. Type I IFNs also promote activation of natural killer (NK) cell cytotoxicity and NK and natural killer T (NKT) cell IFN-γ production. Gamma delta (γδ) T cells, which have both innate and adaptive properties, are also thought to play a protective role against HCMV infection [9,10].

Despite the activation of an antiviral response, HCMV establishes latent infection by residing in monocyte precursors and tissue stromal cells which undergo constant host immune surveillance to prevent reactivation of infection [11]. While it is the adaptive immune system that is responsible for immune surveillance and long-term control of viral replication to maintain a state of latent infection [11,12], early in infection the innate immune system and NK cell activation are also thought to play a critical role [7,9,10]. Cytomegalovirus has co-evolved with its vertebrate hosts for over 100 million years to become a master manipulator of host immunity in order to maintain long-term asymptomatic latent infection [13,14]. These involve multiple immune escape strategies to avoid both innate and adaptive immunity [14], resulting in what has been termed “mutually assured survival”. This involves the expression of host protein homologues including chemokines and a range of immunomodulatory proteins unique to HCMV, including proteins that can evade type I IFN signalling [15] and mechanisms to avoid NK cell-mediated effector functions [7,16]. The CMV immediate-early (IE) gene expresses nuclear phosphoproteins that act as transcriptional regulators involved in evasion of innate immunity and reactivation from latency [17]. The immunosuppressive cytokine IL-10 also plays a role in latent infection, and MCMV expresses an IL-10 orthologue gene during the latent phase [18]. Unfortunately, much of the MCMV evidence cannot be extrapolated to humans since it derives from murine models of infection which have unique immune-modulatory genes that are not present in HCMV [19].

We now understand that the innate immune system has memory, a process that has been called “trained immunity,” which operates largely via epigenetic effects [20]. It is therefore not surprising that HCMV has been shown to have long-lasting adaptive effects on innate immunity, including epigenetic modification of NK cells [21]. This NK cell training impacts the innate response not only to CMV [22] but potentially to other immune challenges. There is a need for research to understand how asymptomatic latent HCMV infection impacts ability to respond to challenge by diverse pathogens and vaccination. This is particularly pertinent to high disease burden settings such as Africa, where HCMV infection rates are almost 100% by one year of age [1] and where multiple vaccinations are administered to HCMV+ infants.

Sex differences in susceptibility and outcomes from many infections are well described, including viruses [23], and females seem to have a greater HCMV seroprevalence in most studies, suggesting greater susceptibility [24]. Immunologically, sex differences have been shown for several aspects of HCMV influences on adaptive immunity, including circulating memory T cell subsets [25] and T cell cytokine profiles [26]. One study hints at sex differences in the impact of HCMV on circulating NK cell frequencies, with males having higher CD56^dim^NK cells (mature NK cells) than females [27], but otherwise, sex-differential effects of HCMV infection on innate immunity have not been described.

While the effects of HCMV on the innate immune system have therefore been characterised in some detail, mostly in animal models, the consequences of asymptomatic latent HCMV infection in early life on innate reactivity to immunological challenge in vivo is not known. It is also not known whether biological sex would have an influence on any observed effects in prepubertal children. We therefore selected a panel of TLR ligands to interrogate the ex vivo innate cytokine response in African infant males and females participating in a vaccine trial [28]. An investigation of the impact of HCMV on TLR reactivity is particularly timely since TLR ligands, particularly TLR7/8 agonists, are being developed as adjuvants for infant vaccines [29]. These will be employed in infants in low-income settings with very high rates of infant HCMV infection.

## 2. Materials and Methods

### 2.1. Study Population

The 9-month old infants in this study were participating in a vaccine trial, the results of which have been published previously [28,30]. They were healthy afebrile male and female children recruited at the Sukuta Health Centre near the coast of The Gambia in West Africa. All study children were well on the day of sample collection and had received all recommended vaccines except for two-thirds of the children who had only received 2 diphtheria-tetanus-whole cell pertussis (DTP) doses as required in the study protocol.

### 2.2. Ethics Statement

The study was approved by the Joint Gambia Government/MRC Ethics Committee (project number SCC1085) and the London School of Hygiene and Tropical Medicine Ethics Committee. Written/thumb-printed informed consent was provided by the parent/guardian of the child in accordance for the Declaration of Helsinki.

### 2.3. Diagnosis of HCMV Infection

Urine samples collected at 9 months of age were tested for the presence of HCMV DNA by a nested polymerase chain reaction (PCR) with a cut off of approximately 25 DNA copies/mL, as described previously [31,32]. Serum HCMV IgG was not analysed since it is unreliable in this age group due to the potential for false-positive results from maternally derived antibody. A negative control known to contain no HCMV was used in every PCR run, and positive controls of 250 and 2500 DNA copies/mL were also included to reduce the likelihood of false-positive diagnoses. Urine was not available at 9 months for all infants in the study cohort, and only those with a urine HCMV result are included.

### 2.4. Whole Blood Cultures and Cytokine Multiplex Analysis

A heparinised whole venous blood sample was taken at 9 months of age prior to administration of the study vaccines. Whole blood was cultured in 100 μL aliquots in 96-well U-bottom plates with a panel of TLR ligands: heat-killed listeria monocytogenes (HKLM) (10^9^ cells/mL) (TLR2 agonist), *Escherichia coli K12* lipopolysaccharide protein S (LPS) (1 μg/mL) (TLR4 agonist), flagellin (10 μg/mL) (TLR5 agonist) and CLO75 (10 μg/mL) (TLR7/8 agonist) (all from InvivoGen, San Diego, CA, USA). The TLR ligand concentrations used were selected as optimal in prior titration assays in infants and are in line with our previous published studies in this age group [33]. RPMI alone was used in the unstimulated control wells. Plates were incubated for 16 h at 37 °C, 5% CO_2_, centrifuged at 2000 rpm for 5 min and 50 μL supernatant collected and stored at −20 °C. The Bio-Plex 200 Suspension Array system (Bio-Rad, Hercules, CA, USA) was used to analyse cytokines according to the manufacturer’s instructions (Bio-Rad, Belgium). A 5-plex array (IL-1β, IL-6, IL-10, IL-12(p70), TNF-α) was used. Innate assays were only performed for those infants in the study with sufficient blood volume and quality, and those with low volume, clotted or contaminated blood samples were excluded.

### 2.5. Statistical Analysis

General linear modelling was used to estimate the geometric mean, geometric standard deviation (GSD), mean difference, 95% confidence interval and *p*-value for each variable for infants with and without HCMV infection. The effects of infant sex were determined by estimating HCMV:sex interactions. Analysis was performed on natural logarithmic transformed cytokine levels and then converted back to natural numbers by exponential transformation; thus, the GSD is a multiplicative factor and not an absolute value. In some cases, there was insufficient blood: it clotted or wells became contaminated; thus, cytokine values were missing for 95 (3.5%) of the potential 2700 wells in the study (108 infants tested for 5 cytokines to 5 conditions (unstimulated and 4 TLR ligands)). These missing data were substituted by multiple imputation (using “mi estimate: glm” Stata syntax, with 50 imputations). Multivariate linear regression modelling was used in order to capture the potential inter-relationships between the immune parameters tested. For this, infant sex, HCMV status and all other cytokine/chemokine values were considered potential covariates for adjustment. Covariates selected by forward stepwise regression had an entry *p*-value of 0.15 and a removal *p*-value of 0.2 and were presented to the statistical model as z-scores of the natural logarithms of the raw values. The z-scores used for the adjustment process were the subject value minus the mean value divided by the standard deviation. The adjusted values therefore control for the responses to all the other TLR agonists. Unadjusted and adjusted *p*-values are shown in the tables. All statistical analyses were performed using Stata MP2 V16.0 (StataCorp, College Station, TX, USA).

## 3. Results

### 3.1. Infant Characteristics

Data from 108 study infants who had both urine HCMV and innate cytokine data at 9 months of age are included in the analysis. Seventy-three (67.6%) of the 108 infants were HCMV-positive (HCMV+) at 9 months of age, and 35 (32.4%) were HCMV-negative (HCMV-) (Table 1). There was an approximately equal number of males and females in the analysis, which included 55 (50.9% of participants) males and 53 (49.1% of participants) females; 66% of males were HCMV+, and 69.1% of females were HCMV+ (Table 1). All samples were collected during the dry season, which runs from November to May in The Gambia. All children were well and afebrile on the day of blood collection; we therefore did not test for any other pathogens such as malaria.

### 3.2. Minimal Effects of HCMV Infection on TLR Ligand Responses When Males and Females Are Analysed Together

When all infants were analysed together, the only TLR agonist/cytokine combination affected by HCMV infection was a lower TNF-α to HKLM stimulation in the HCMV infected as compared to the uninfected group (*p* = 0.04 unadjusted, 0.066 adjusted) (Table 2). IL-1β was also lower in unstimulated control wells in infants infected with HCMV as compared to the uninfected infants (*p* = 0.81 unadjusted, 0.022 adjusted) (Table 3). IL-6, IL-12 (p70), and IL-10 responses in the TLR ligand cultures were not affected by HCMV status when all infants were analysed together (Table 4, Table 5 and Table 6). The TNF-α:IL-10 pro-:anti-inflammatory cytokine ratio was also not affected by HCMV status when analysing all infants combined (Table 7).

### 3.3. Evidence for Effects of HCMV Infection on TLR Ligand Responses in Females but Not Males

TNF-α and TNF-α:IL-10 responses to the TLR2 ligand HKLM were lower in HCMV+ females as compared to the HCMV- females (TNF-α: unadjusted *p* = 0.021, adjusted *p* = 0.022; TNF-α:IL-10: unadjusted *p* = 0.038, adjusted *p* = 0.021) (Figure 1A), whereas there was no apparent effect of HCMV infection in females or males on TNF-α responses to the other three TLR ligands—LPS, flagellin or CLO75 (Table 2 and Table 7).

HCMV+ females had lower IL-6 reactivity to the TLR7/8 ligand CLO75 as compared to their HCMV- counterparts (unadjusted *p* = 0.014, adjusted *p* = 0.008), while HCMV+ males had greater CLO75-stimulated IL-6 responses as compared to the HCMV- males (unadjusted *p* = 0.15, adjusted *p* = 0.018) (Figure 1B, Table 4). A comparison of the effect of HCMV effect in males and females showed a highly significant difference between the sexes with respect to CLO75 IL-6 responses (unadjusted *p* = 0.005, adjusted *p* < 0.001) (Figure 1B, Table 4).

By contrast, IL-10 responses to CLO75 were higher in HCMV+ females as compared to HCMV- females (unadjusted *p* = 0.47, adjusted *p* = 0.005), while there was no such effect of HCMV status in the male group (Figure 1C, Table 5). The HCMV effect was significantly different between males and females for CLO75-stimulated IL-10 production (*p* = 0.016) (Figure 1C, Table 5). IL-10 responses to the TLR4 ligand LPS were also significantly different when comparing the HCMV effect in males and females (unadjusted *p* = 0.040, adjusted *p* = 0.034) (Figure 1D). This is because HCMV+ females had higher IL-10 than the HCMV- females, while HCMV+ males had the reciprocal effect of lower IL-10 than their HCMV- counterparts, albeit the differences were not significant in the separate sex analyses (Figure 1D, Table 5).

The only TLR ligand responses not affected by HCMV status were cytokine responses to the TLR5 agonist flagellin. Furthermore, HCMV status had no effect on IL-1β or IL-12 production to the panel of TLR ligands.

## 4. Discussion

The effect of HCMV infection on innate reactivity has not been investigated in infants in areas of the world where infection rates reach almost 100% by one year of age [1]. Furthermore, the possibility that effects might differ in the sexes has not been investigated. Herein, we show that in the non-sex-stratified analysis, HCMV infection had little impact on in vitro innate cytokine responses to a panel consisting of TLR2, 4, 5 and 7/8 ligands, except for reduced TNF-α reactivity to TLR2 stimulation with HKLM.

Sex differences in susceptibility to infections and responses to vaccination are evident from birth, with males being the more susceptible to a range of infections and more likely to succumb to sepsis, while females have more robust antibody responses to many vaccines [23,34,35]. On the few occasions that it has been examined, infant males have been shown to have greater innate immune responses to vaccination as compared to females [35]. The potential reasons for these immunological sex differences include sex differences in circulating sex hormone levels, sex differences in the microbiota and sex differences in the expression of the multiple immune response genes expressed on the X chromosome [23,34,35]. When males and females were analysed separately in our study, we found that HCMV+ females had reduced TNF-α and TNF-α:IL-10 to HKLM stimulation, but there was no such effect of HCMV infection in males. Females also had reduced IL-6 and increased IL-10 to CLO75 stimulation. By contrast, HCMV+ males had increased IL-6 to CLO75 stimulation. TNF-α and IL-6 are classic pro-inflammatory cytokines [36], while IL-10 has suppressive immunoregulatory properties [37]. Therefore, our results suggest that HCMV infection suppresses innate reactivity to TLR2 and TLR7/8 stimulation in infant females but not in males. This could compromise female reactivity to pathogens recognised via TLR2 and TLR7/8 receptors, which include a range of Gram-positive and Gram-negative bacteria, viruses, yeasts, mycoplasma and parasites [38,39].

These findings have potentially important implications given the high incidence of infant HCMV infection in low-income settings. If HCMV infection suppresses innate immunity, particularly in females, this could alter female susceptibility to the plethora of infections to which they are exposed in early life. Furthermore, vaccine immunogenicity relies on an early innate response, which is intrinsically linked to the adaptive immune response [40]. Suppressed innate immunity in HCMV+ females could therefore result in poorer innate and adaptive immune responses to vaccination. Indeed, our recent study found that HCMV+ females had lower tetanus toxoid and pertactin IgG titres following diphtheria–tetanus–whole-cell-pertussis immunisation as compared to their HCMV- counterparts, although only on unadjusted analysis [41]. Having said that, most infants achieved seroprotective levels regardless of HCMV status in this study, suggesting limited clinical impact of HCMV infection [41]. In the same study, HCMV+ infants had suppressed overall T cell immunity, as evidenced by lower cytokine responses to stimulation with anti-CD3/anti-CD28 [41]. Overall, this suggests that HCMV infection can suppress both innate and adaptive immunity in infants in high-HCMV-incidence settings, consistent with its well-described evolutionary mechanisms aimed at avoiding host immunity.

Several limitations of this study should be borne in mind. The relatively small number of infants in the HCMV- groups reduces the power of the study. Another issue is that some of the HCMV- infants may have been infected earlier in life but no longer shedding the virus in their urine and thus classified as HCMV-. However, previous studies in Gambian infants show that those infected in the first year of life continue to shed virus in urine for many months [1,42], so we do not think that many infants would have been misclassified. Future longitudinal studies would help confirm whether the phenotype we describe is specific to active shedders or all infants who have acquired HCMV infection. The whole blood assay (WBA) used in this study is considered a more physiological assay, representative of true in vivo reactivity, compared to using separated peripheral blood mononuclear cells or other purified cell populations. A variety of cells in whole blood contribute to the cytokine responses to TLR stimulation, including activated monocytes, macrophages, dendritic cells, NK cells, NK T cell, γδ T cells and neutrophils. Therefore, a limitation of this approach is that we do not know precisely which cells are producing the cytokines measured.

It is interesting that responses to the TLR7/8 ligand CLO75 are particularly affected by co-existing HCMV infection in our study. Newborns have a robust response to this TLR ligand from birth [33,43], and as a result, it is being developed as an adjuvant for infant vaccines, particularly for neonates [29,44,45]. Our results suggest that HCMV+ infant females may have impaired reactivity to this adjuvant approach.

## 5. Conclusions

In summary, we show for the first time that early-life HCMV infection can impact innate immunity, in infant females in particular, with potential implications for their ability to respond to infections and vaccines. The clinical implications are uncertain, but further contribute to our understanding of the well-described sex differences in disease susceptibility and vaccine responses in the prepubertal period and lend further weight to the arguments for developing an effective HCMV vaccine.

## Figures and Tables

**Figure 1 vaccines-08-00407-f001:**
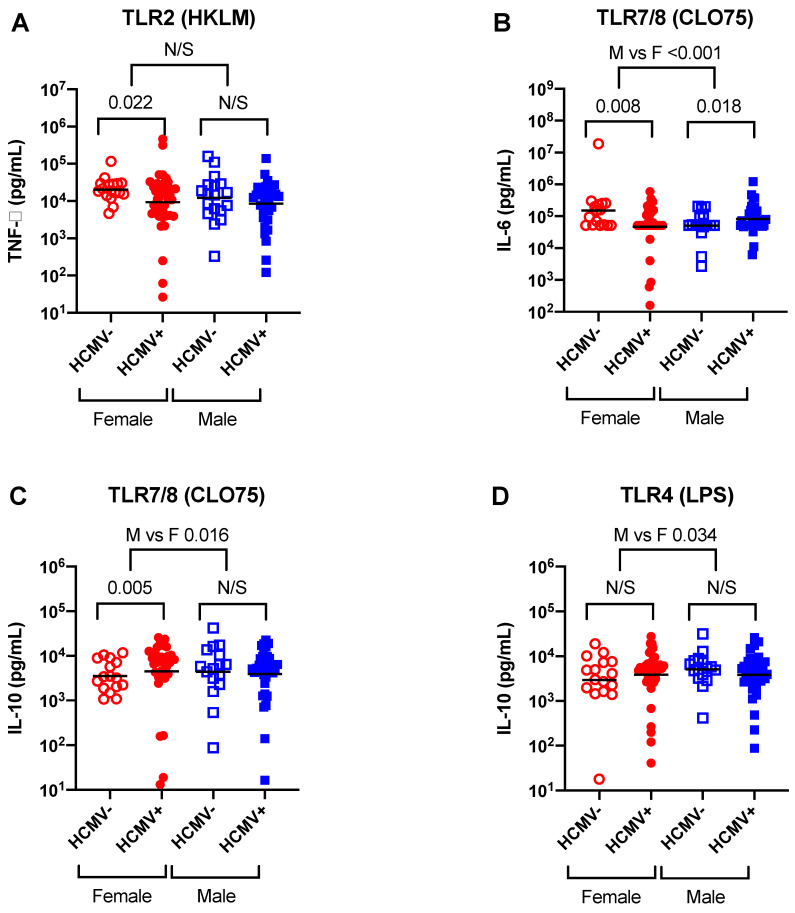
The significant sex-differential cytokine responses in the study. Whole heparinized blood was stimulated for 16 h with TLR ligands, and cytokine concentrations were measured in culture supernatants by multiplex assay. (**A**). HCMV+ females had lower TNF-α responses to TLR2 (HKLM) stimulation than HCMV- females. (**B**). HCMV+ females had lower IL-6 responses to TLR7/8 (CLO75) stimulation than HCMV- females, while males had the opposite pattern of higher responses in the HCMV+ group. There was a significant difference in IL-6 responses to CLO75 between males and females. (**C**). HCMV+ females had greater IL-10 production to CLO75 than HCMV- females, whereas HCMV+ and HCMV- males had comparable CLO75-stimulated IL-10. There was a significant difference in IL-10 responses to CLO75 between males and females. (**D**). There was no significant difference in IL-10 responses to TLR4 (LPS) stimulation between HCMV+ and HCMV- infants for males or females, but comparing male and female responses did show a significant difference. The individual symbols equal individual infants and the bar shows the geometric mean value in pg/mL. N/S = not significant, M = males and F = females.

**Table 1 vaccines-08-00407-t001:** Infant numbers according to sex and HCMV status.

Group	HCMV+	HCMV-	Total
All Infants	73 (67.6%)	35 (32.4%)	108
Males	35 (66.0%)	18 (34.0%)	53
Females	38 (69.1%)	17 (30.9%)	55

Number of infants shown with % of total in brackets.

**Table 2 vaccines-08-00407-t002:** Effect of HCMV infection on TLR ligand-stimulated TNF-α responses by infant sex.

TNF-α	HCMV−		HCMV+		Comparison (Unadjusted)			Comparison (Adjusted)		
	N	Geo Mean (GSD)	N	Geo Mean (GSD)	Δ	95% CI	*p*-Value	Δ	95% CI	*p*-Value
**Unstimulated**										
ALL	34	73 (9.0)	73	97 (9.5)	25	(−33 to 165)	0.52	426	(−88 to 1144)	0.12
Male	17	139 (9.7)	35	111 (9.5)	−27	(−109 to 278)	0.75	588	(−56 to 1581)	0.08
Female	17	38 (7.2)	38	89 (9.8)	51	(−10 to 238)	0.15	272	(−1874 to 1503)	0.54
M vs. F					−78	(−170 to 446)	0.23	316	(−585 to 2092)	0.51
**TLR2 (HKLM)**										
ALL	35	15,529 (3.2)	73	8818 (5.3)	−6711	(−10,386 to −409)	**0.040**	−1369	(−2414 to 107)	0.066
Male	18	12,050 (4.2)	35	7843 (4.4)	−4207	(−8630 to 5937)	0.31	−338	(−1712 to 1966)	0.72
Female	17	20,311 (2.0)	38	9614 (6.1)	−10,698	(−15,214 to −2178)	**0.021**	−2738	(−7615 to −497)	**0.022**
M vs. F					6490	(−1406 to 20,944)	0.55	2400	(−679 to 5880)	0.21
**TLR4 (LPS)**										
ALL	32	6349 (5.0)	72	4661 (5.4)	−1688	(−3978 to 2816)	0.37	−181	(−1108 to 1254)	0.76
Male	16	6115 (4.2)	34	5027 (4.5)	−1088	(−4019 to 5941)	0.66	27	(−1150 to 2222)	0.97
Female	16	6591 (6.2)	38	4429 (6.0)	−2161	(−4996 to 5709)	0.45	−396	(−3837 to 1839)	0.66
M vs. F					1073	(−2644 to 15,336)	0.77	422	(−1047 to 3909)	0.74
**TLR5 (Flagellin)**										
ALL	33	5245 (4.3)	69	4405 (6.1)	−840	(−2948 to 3202)	0.60	−239	(−945 to 875)	0.62
Male	17	5137 (4.6)	34	4326 (5.4)	−811	(−3429 to 5824)	0.72	151	(−749 to 1848)	0.80
Female	16	5362 (4.2)	35	4462 (6.8)	−900	(−3553 to 5645)	0.69	−690	(−3257 to 895)	0.32
M vs. F					89	(−3053 to 11,572)	0.99	842	(−247 to 3360)	0.37
**TLR7/8 (CLO75)**										
ALL	32	3960 (3.5)	68	3523 (6.7)	−438	(−2072 to 2610)	0.71	−130	(−798 to 830)	0.76
Male	16	3420 (4.6)	34	2678 (4.8)	−742	(−2358 to 3331)	0.60	−164	(−916 to 1123)	0.75
Female	16	4586 (2.5)	34	4319 (8.3)	−267	(−2628 to 4943)	0.88	−74	(−2341 to 1649)	0.91
M vs. F					−475	(−2360 to 5890)	0.77	−91	(−1054 to 1978)	0.88

TNF-α levels were analysed in culture supernatants from whole blood stimulated for 16 h with toll-like receptor (TLR) ligands. The table shows geometric mean (geo mean) plus geometric standard deviation (GSD) cytokine levels in pg/mL. The mean difference (Δ; 95% confidence intervals; *p*-values) between those geometric means were estimated using general linear modelling (unadjusted and adjusted for covariates) with missing data substituted by multiple imputation. The difference between the effect of Human cytomegalovirus (HCMV) in males compared with females was estimated (M vs. F). N = number of infants analysed for each TLR ligand. Bold are the significant effects.

**Table 3 vaccines-08-00407-t003:** Effect of HCMV infection on TLR ligand-stimulated IL-1β responses by infant sex.

IL-1β	HCMV−		HCMV+		Comparison (Unadjusted)			Comparison (Adjusted)		
	N	Geo Mean (GSD)	N	Geo Mean (GSD)	Δ	95%CI	*p*-Value	Δ	95%CI	*p*-Value
**Unstimulated**										
ALL	34	83 (13.2)	73	73 (11.8)	−10	(−57 to 120)	0.81	−1223	(−1939 to −212)	**0.022**
Male	17	212 (14.4)	35	86 (14.4)	−126	(−194 to 194)	0.25	−1460	(−2463 to 329)	0.094
Female	17	32 (8.7)	38	64 (10.4)	32	(−13 to 187)	0.27	−971	(−4292 to 415)	0.15
M vs. F					−159	(−232 to 376)	0.11	−489	(−1693 to 2056)	0.65
**TLR2 (HKLM)**										
ALL	35	14,290 (3.4)	73	10,137 (7.0)	−4154	(−8757 to 4280)	0.27	97	(−1256 to 2168)	0.91
Male	18	13,556 (3.7)	35	8436 (4.9)	−5120	(−9800 to 5390)	0.25	−379	(−1729 to 1869)	0.68
Female	17	15,111 (3.3)	38	11,646 (8.9)	−3464	(−10,213 to 12,584)	0.56	630	(−5053 to 4807)	0.65
M vs. F					−1655	(−7513 to 17,510)	0.72	−1009	(−2943 to 3511)	0.54
**TLR4 (LPS)**										
ALL	32	5668 (5.7)	72	4038 (6.7)	−1630	(−3744 to 2809)	0.37	−275	(−1063 to 1021)	0.61
Male	16	5280 (3.2)	34	4519 (4.9)	−761	(−3252 to 4789)	0.70	137	(−830 to 1985)	0.83
Female	16	6084 (9.0)	38	3737 (8.3)	−2348	(−4987 to 6637)	0.44	−717	(−4625 to 1314)	0.39
M vs. F					1587	(−1887 to 16,599)	0.66	854	(−370 to 3944)	0.43
**TLR5 (Flagellin)**										
ALL	33	5063 (4.1)	69	4024 (4.4)	−1039	(−2832 to 2194)	0.45	−491	(−1146 to 468)	0.27
Male	17	6032 (4.0)	34	3994 (3.4)	−2037	(−4206 to 2706)	0.30	−897	(−1665 to 345)	0.13
Female	16	4204 (4.2)	35	4046 (5.2)	−158	(−2487 to 5332)	0.93	−132	(−2266 to 1409)	0.83
M vs. F					−1880	(−4623 to 6879)	0.53	−765	(−1768 to 1240)	0.38
**TLR7/8 (CLO75)**										
ALL	32	3940 (4.2)	68	3014 (6.5)	−926	(−2389 to 1917)	0.43	−348	(−812 to 307)	0.26
Male	16	3536 (6.0)	34	2898 (4.0)	−638	(−2456 to 4246)	0.69	−228	(−904 to 884)	0.63
Female	16	4390 (2.8)	34	3105 (8.9)	−1284	(−3053 to 2823)	0.42	−468	(−2054 to 481)	0.28
M vs. F					647	(−1460 to 8365)	0.82	240	(−638 to 2029)	0.69

IL-1β levels were analysed in culture supernatants from whole blood stimulated for 16 h with TLR ligands. The table shows geometric mean (geo mean) plus geometric standard deviation (GSD) cytokine levels in pg/mL. The mean difference (Δ; 95% confidence intervals; *p*-values) between those geometric means was estimated using general linear modelling (unadjusted and adjusted for covariates) with missing data substituted by multiple imputation. The difference between the effect of HCMV in males compared with females was estimated (M vs. F). N = number of infants analysed for each TLR ligand.

**Table 4 vaccines-08-00407-t004:** Effect of HCMV infection on TLR ligand-stimulated IL-6 responses by infant sex.

IL-6	HCMV−		HCMV+		Comparison (Unadjusted)			Comparison (Adjusted)		
	N	Geo Mean (GSD)	N	Geo Mean (GSD)	Δ	95%CI	*p*-Value	Δ	95%CI	*p*-Value
**Unstimulated**										
ALL	34	2188 (10.0)	73	2480 (7.2)	293	(−1167 to 3839)	0.78	6906	(−2431 to 21,524)	0.17
Male	17	5791 (8.5)	35	3312 (7.3)	−2480	(−4810 to 5388)	0.37	2717	(−11,543 to 30,714)	0.78
Female	17	826 (8.0)	38	2027 (7.0)	1201	(−166 to 5401)	0.12	8854	(−11,251 to 26,477)	0.091
M vs. F					−3681	(−6360 to 10,339)	0.085	−6137	(−23,124 to 34,746)	0.37
**TLR2 (HKLM)**										
ALL	35	99,711 (2.6)	73	73,393 (3.2)	−26,317	(−51,057 to 11,003)	0.14	−8868	(−26,981 to 17,855)	0.46
Male	18	93,368 (2.9)	35	88,146 (3.1)	−5221	(−45,992 to 70,635)	0.86	6634	(−22,721 to 58,030)	0.72
Female	17	10,6898 (2.3)	38	64,113 (3.3)	−42,785	(−69,160 to 2024)	0.06	−23,230	(−96,836 to 10,025)	0.14
M vs. F					37,564	(−11,617 to 14,8820)	0.28	29,863	(−6245 to 10,6282)	0.20
**TLR4 (LPS)**										
ALL	32	98,179 (4.4)	72	61,846 (2.9)	−36,334	(−62,872 to 10,152)	0.11	−17,692	(−35,345 to 10,443)	0.18
Male	16	11,4889 (2.7)	34	75,326 (2.4)	−39,563	(−72,343 to 18,475)	0.15	−18,329	(−42,013 to 23,017)	0.32
Female	16	83,899 (6.3)	38	54,144 (3.2)	−29,755	(−62,884 to 55,601)	0.36	−17,274	(−10,0509 to 25,908)	0.34
M vs. F					−9808	(−60,196 to 14,2396)	0.98	−1055	(−33,561 to 77,520)	0.89
**TLR5 (Flagellin)**										
ALL	33	71,286 (3.6)	69	69,451 (2.8)	−1836	(−28,974 to 42,709)	0.92	861	(−16,017 to 27,511)	0.94
Male	17	77,027 (3.1)	34	72,256 (2.7)	−4771	(−38,656 to 59,038)	0.84	−73	(−20,553 to 35,652)	1.00
Female	16	65,655 (4.3)	35	67,485 (2.9)	1830	(−34,427 to 80,182)	0.94	1670	(−58,715 to 47,904)	0.92
M vs. F					−6602	(−52,201 to 11,7002)	0.86	−1743	(−30,217 to 68,278)	0.93
**TLR7/8 (CLO75)**										
ALL	32	87,048 (4.1)	68	61,996 (4.3)	−25,052	(−52,802 to 25,183)	0.26	−8595	(−26,326 to 20,587)	0.49
Male	16	50,778 (3.2)	34	82,605 (2.8)	31,827	(−8476 to 11,0530)	0.15	28,847	(3689 to 72,477)	**0.018**
Female	16	14,9225 (4.3)	34	50,082 (5.3)	−99,143	(−12,8262 to −29,577)	**0.014**	−57,211	(−15,7017 to −20,296)	**0.008**
M vs. F					13,0971	(75,908 to 29,6114)	**0.005**	86,058	(50,886 to 17,2231)	**<0.001**

IL-6 levels were analysed in culture supernatants from whole blood stimulated for 16 h with TLR ligands. The table shows geometric mean (geo mean) plus geometric standard deviation (GSD) cytokine levels in pg/mL. The mean difference (Δ; 95% confidence intervals; *p*-values) between those geometric means were estimated using general linear modelling (unadjusted and adjusted for covariates) with missing data substituted by multiple imputation. The difference between the effect of HCMV in males compared with females was estimated (M vs. F). N = number of infants analysed for each TLR ligand. Bold are the significant effects.

**Table 5 vaccines-08-00407-t005:** Effect of HCMV infection TLR on ligand-stimulated IL-12(p70) responses by infant sex.

IL-12	HCMV−		HCMV+		Comparison (Unadjusted)			Comparison (Adjusted)		
	N	Geo Mean (GSD)	N	Geo Mean (GSD)	Δ	95% CI	*p*-Value	Δ	95% CI	*p*-Value
**Unstimulated**										
ALL	34	25 (3.9)	73	25 (6.1)	0	(−11 to 22)	0.95	−19	(−60 to 50)	0.52
Male	17	40 (3.1)	35	27 (6.5)	−13	(−28 to 22)	0.35	−51	(−101 to 48)	0.24
Female	17	16 (4.3)	38	25 (6.0)	9	(−5 to 42)	0.30	5	(−166 to 113)	0.89
M vs. F					−22	(−41 to 40)	0.17	−56	(−120 to 114)	0.36
**TLR2 (HKLM)**										
ALL	35	117 (2.1)	73	116 (2.7)	−1	(−35 to 45)	0.95	7	(−9 to 28)	0.41
Male	18	121 (2.2)	35	115 (2.1)	−6	(−47 to 56)	0.82	11	(−14 to 45)	0.44
Female	17	113 (2.1)	38	117 (3.2)	3	(−42 to 78)	0.91	4	(−71 to 35)	0.74
M vs. F					−9	(−65 to 98)	0.81	6	(−26 to 59)	0.75
**TLR4 (LPS)**										
ALL	32	105 (2.2)	72	108 (3.0)	3	(−31 to 51)	0.90	5	(−15 to 32)	0.66
Male	16	100 (2.2)	34	115 (2.5)	14	(−30 to 87)	0.59	17	(−9 to 54)	0.22
Female	16	110 (2.3)	38	103 (3.3)	−7	(−50 to 66)	0.81	−7	(−77 to 34)	0.70
M vs. F					21	(−38 to 144)	0.59	24	(−12 to 86)	0.27
**TLR5 (Flagellin)**										
ALL	33	92 (2.5)	69	112 (2.6)	19	(−17 to 72)	0.34	7	(−9 to 28)	0.44
Male	17	105 (2.5)	34	110 (3.2)	5	(−45 to 95)	0.88	2	(−21 to 38)	0.88
Female	16	81 (2.5)	35	113 (2.3)	32	(−13 to 107)	0.20	11	(−25 to 37)	0.28
M vs. F					−27	(−87 to 104)	0.47	−9	(−36 to 38)	0.58
**TLR7/8 (CLO75)**										
ALL	32	95 (3.4)	68	123 (3.5)	28	(−21 to 111)	0.32	13	(−11 to 47)	0.32
Male	16	100 (3.4)	34	112 (2.6)	12	(−44 to 120)	0.75	28	(−6 to 82)	0.12
Female	16	89 (3.5)	34	132 (4.3)	42	(−27 to 189)	0.31	−3	(−78 to 44)	0.89
M vs. F					−31	(−102 to 165)	0.59	31	(−13 to 113)	0.22

IL-12(p70) levels were analysed in culture supernatants from whole blood stimulated for 16 h with TLR ligands. The table shows geometric mean (geo mean) plus geometric standard deviation (GSD) cytokine levels in pg/mL. The mean difference (Δ; 95% confidence intervals; *p*-values) between those geometric means were estimated using general linear modelling (unadjusted and adjusted for covariates) with missing data substituted by multiple imputation. The difference between the effect of HCMV in males compared with females was estimated (M vs. F). N= number of infants analysed for each TLR ligand.

**Table 6 vaccines-08-00407-t006:** Effect of HCMV infection on TLR ligand-stimulated IL-10 responses by infant sex.

IL-10	HCMV−		HCMV+		Comparison (Unadjusted)			Comparison (Adjusted)		
	N	Geo Mean (GSD)	N	Geo Mean (GSD)	Δ	95%CI	*p*-Value	Δ	95%CI	*p*-Value
**Unstimulated**										
ALL	34	76 (8.0)	73	96 (8.4)	20	(−35 to 148)	0.59	136	(−350 to 875)	0.64
Male	17	149 (9.1)	35	108 (9.2)	−40	(−119 to 245)	0.63	32	(−596 to 1172)	0.94
Female	17	39 (5.8)	38	89 (8.1)	49	(−7 to 208)	0.12	221	(−1363 to 1368)	0.58
M vs. F					−90	(−177 to 365)	0.18	−189	(−988 to 1675)	0.73
**TLR2 (HKLM)**										
ALL	35	4295 (3.3)	73	3409 (4.3)	−886	(−2258 to 1409)	0.38	98	(−243 to 563)	0.62
Male	18	4786 (3.8)	35	3366 (4.6)	−1420	(−3273 to 2699)	0.39	−134	(−530 to 437)	0.60
Female	17	3830 (2.8)	38	3441 (4.3)	−389	(−2035 to 2768)	0.75	283	(−902 to 1120)	0.33
M vs. F					−1031	(−3197 to 5038)	0.64	−418	(−1024 to 728)	0.27
**TLR4 (LPS)**										
ALL	32	3820 (3.7)	72	3593 (4.0)	−227	(−1744 to 2398)	0.83	299	(−348 to 1199)	0.41
Male	16	5016 (2.5)	34	3775 (3.6)	−1241	(−3007 to 2078)	0.38	−533	(−1260 to 553)	0.29
Female	16	2910 (4.8)	38	3475 (4.5)	566	(−1455 to 5393)	0.69	860	(−859 to 2297)	0.062
M vs. F					−1807	(−4294 to 5484)	0.40	−1393	(−2409 to −495)	**0.034**
**TLR5 (Flagellin)**										
ALL	33	4409 (3.4)	69	5412 (2.6)	1003	(−1040 to 4285)	0.40	627	(−226 to 1753)	0.17
Male	17	4901 (3.4)	34	5327 (2.5)	425	(−2125 to 5319)	0.80	590	(−637 to 2483)	0.40
Female	16	3940 (3.6)	35	5474 (2.8)	1534	(−1180 to 6917)	0.35	661	(−1569 to 2214)	0.27
M vs. F					−1109	(−4366 to 7274)	0.61	−71	(−1593 to 2630)	0.94
**TLR7/8 (CLO75)**										
ALL	32	3941 (3.2)	68	4198 (5.1)	257	(−1537 to 3391)	0.82	350	(−266 to 1148)	0.29
Male	16	4382 (4.3)	34	3786 (4.6)	−596	(−2826 to 4825)	0.75	−580	(−1416 to 684)	0.32
Female	16	3544 (2.2)	34	4534 (5.6)	990	(−1201 to 5231)	0.47	1069	(337 to 2195)	**0.005**
M vs. F					−1586	(−4121 to 6083)	0.49	−1649	(−2704 to −194)	**0.016**

IL-10 levels were analysed in culture supernatants from whole blood stimulated for 16 h with TLR ligands. The table shows geometric mean (geo mean) plus geometric standard deviation (GSD) cytokine levels in pg/mL. The mean difference (Δ; 95% confidence intervals; *p*-Values) between those geometric means were estimated using general linear modelling (unadjusted and adjusted for covariates) with missing data substituted by multiple imputation. The difference between the effect of HCMV in males compared with females was estimated (M vs. F). N = number of infants analysed for each TLR ligand. Bold are the significant effects.

**Table 7 vaccines-08-00407-t007:** Effect of HCMV infection TLR ligand-stimulated TNF-α:IL-10 ratio in culture supernatants by infant sex.

TNF:IL-10	HCMV−		HCMV+		Comparison (Unadjusted)			Comparison (Adjusted)		
	N	Geo Mean (GSD)	N	Geo Mean (GSD)	Δ	95% CI	*p*-Value	Δ	95% CI	*p*-Value
**Unstimulated**										
ALL	34	0.95 (3.0)	73	0.97 (3.0)	0.02	(−0.33 to 0.56)	0.94	0.18	(−0.08 to 0.54)	0.19
Male	17	0.93 (2.8)	35	1.03 (2.7)	0.10	(−0.36 to 0.92)	0.75	0.33	(−0.02 to 0.86)	0.064
Female	17	0.97 (3.3)	38	0.93 (3.3)	−0.04	(−0.49 to 0.82)	0.89	0.04	(−1.08 to 0.60)	0.86
M vs. F					0.14	(−0.46 to 1.59)	0.75	0.29	(−0.18 to 1.18)	0.26
**TLR2 (HKLM)**										
ALL	35	3.62 (2.8)	73	2.57 (3.1)	−1.04	(−1.93 to 0.32)	0.12	−0.74	(−1.29 to 0.05)	0.063
Male	18	2.52 (2.5)	35	2.33 (2.5)	−0.19	(−1.13 to 1.39)	0.77	−0.18	(−0.90 to 1.04)	0.72
Female	17	5.30 (2.8)	38	2.77 (3.6)	−2.53	(−3.80 to −0.19)	**0.038**	−1.48	(−4.07 to −0.27)	**0.021**
M vs. F					2.34	(−1.06 to 5.20)	0.16	1.30	(−0.38 to 3.17)	0.20
**TLR4 (LPS)**										
ALL	32	1.66 (3.3)	72	1.29 (3.4)	−0.37	(−0.88 to 0.46)	0.32	−0.09	(−0.58 to 0.66)	0.77
Male	16	1.22 (3.0)	34	1.33 (3.0)	0.11	(−0.53 to 1.35)	0.79	0.03	(−0.58 to 1.17)	0.95
Female	16	2.27 (3.4)	38	1.26 (3.8)	−1.00	(−1.64 to 0.29)	0.10	−0.22	(−2.03 to 0.95)	0.64
M vs. F					1.12	(−0.29 to 3.27)	0.17	0.25	(−0.52 to 2.07)	0.70
**TLR5 (Flagellin)**										
ALL	33	1.19 (3.3)	69	0.82 (3.9)	−0.37	(−0.70 to 0.18)	0.16	−0.12	(−0.50 to 0.47)	0.63
Male	17	1.05 (4.2)	34	0.81 (3.7)	−0.24	(−0.69 to 0.78)	0.54	0.08	(−0.39 to 0.98)	0.79
Female	16	1.36 (2.5)	35	0.83 (4.1)	−0.53	(−0.91 to 0.16)	0.11	−0.36	(−1.74 to 0.50)	0.34
M vs. F					0.30	(−0.22 to 1.73)	0.64	0.44	(−0.14 to 1.78)	0.38
**TLR7/8 (CLO75)**										
ALL	32	1.00 (2.7)	68	0.84 (3.2)	−0.16	(−0.46 to 0.30)	0.43	−0.06	(−0.42 to 0.45)	0.78
Male	16	0.78 (2.9)	34	0.71 (2.8)	−0.07	(−0.40 to 0.55)	0.76	−0.09	(−0.48 to 0.60)	0.75
Female	16	1.29 (2.4)	34	0.96 (3.5)	−0.33	(−0.75 to 0.41)	0.31	−0.02	(−1.24 to 0.92)	0.95
M vs. F					0.26	(−0.15 to 1.20)	0.65	−0.06	(−0.58 to 1.05)	0.86

TNF:IL-10 ratios were analysed in culture supernatants from whole blood stimulated for 16 h with TLR ligands. The table shows geometric mean (geo mean) plus geometric standard deviation (GSD) cytokine levels in pg/m. The mean difference (Δ; 95% confidence intervals; *p*-values) between those geometric means were estimated using general linear modelling (unadjusted and adjusted for covariates) with missing data substituted by multiple imputation. The difference between the effect of HCMV in males compared with females was estimated (M vs. F). N = number of infants analysed for each TLR ligand. Bold are the significant effects.

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
