# Peer review of "Sex-Differential Impact of Human Cytomegalovirus Infection on In Vitro Reactivity to Toll-Like Receptor 2, 4 and 7/8 Stimulation in Gambian Infants"

_vaccines, 2020, doi:10.3390/vaccines8030407_

Round 1

Reviewer 1 Report

It has been a topic of significant interest whether CMV serostatus influences vaccine responses. This is usually pursued from the context of immune exhaustion and whether CMV over the course of an individual’s lifetime weakens new immune responses. In this very interesting study the authors begin with a different supposition, that CMV infection a priori influences subsequent immune responses to vaccines by manipulating innate immunity. To test this, they utilize blood responses to multiple toll like receptors in children defined as CMV-positive or negative.

Major concerns

  1. Because biological studies with human cells are fraught with challenges, occasionally expts don’t go as planned. This is alluded to by the authors in the statistical section, specifically in describing how missing data from failed wells were managed with imputation. Because the sample sizes for the CMV- children are small, and especially so when they are divided by sex, this makes the potential effect of well failure in these subgroups a challenge to control. To increase confidence in the differences shown, it would be helpful to know if such failures were randomly scattered through all groups (CMV+/- males, CMV+/- females). Asked simply, did more failures occur in one subset or another?
  2. While I am picking on subsets, the CMV- subsets are rather small and this limits the power of the study. Acknowledgement of this is lacking.
  3. Because urine DNA were used to define CMV positivity, and not IgG, the identification of truly CMV+ patients is muddled. Urine CMV DNA suggests to this reviewer that the virus could be active in those children causing shedding, and not necessarily latent. While I understand that IgG isn’t ideal for these infants either, the strategy employed allows inclusion of CMV+ children that are not actively shedding in the CMV-negative group. Thus I am not completely convinced that all CMV+ patients are in the CMV+ group, and conversely that not all infants in the CMV- group are actually negative. This leaves the comparison as CMV-negative (includes true negative and CMV-latent not actively shedding) to CMV+ (actively infected shedding). While one might first be excited that despite inclusion of CMV+ children (latent not shedding) in the CMV-negative group there were still detectable differences, a different interpretation (detailed below) is that after active infection CMV+ children responses to TLR revert back to those of CMV-negative children.
  4. Finally, to piggyback on #3, if CMV shedding in the DNA is indicative of active and not latent infection, then evaluating these children at later time points should give a more accurate indication of the impact of latent CMV on immune responses to vaccines earlier in life. This obviously would require significant time and resources and while not necessary for this manuscript, should be considered for future study. Nevertheless, the conclusions drawn here need to at least acknowledge the fact that many (if not all) of the CMV+ children are actively infected, and that further study is required to confirm whether the phenotype observed here is short term or long term. In other words, will the phenotype persist during latency?

    Minor concerns

    1. P2 Row 68 techniques should be changed to mechanisms

Author Response

Major concerns

  1. Because biological studies with human cells are fraught with challenges, occasionally expts don’t go as planned. This is alluded to by the authors in the statistical section, specifically in describing how missing data from failed wells were managed with imputation. Because the sample sizes for the CMV- children are small, and especially so when they are divided by sex, this makes the potential effect of well failure in these subgroups a challenge to control. To increase confidence in the differences shown, it would be helpful to know if such failures were randomly scattered through all groups (CMV+/- males, CMV+/- females). Asked simply, did more failures occur in one subset or another?

    Response: For the reasons alluded to in the question we did not get data for a small number of infants for all 5 TLR ligands. The data for HKLM was complete for all 108 infants, 107 infants for unstimulated control, 104 infants for LPS, 102 for flagellin and 100 for CLO75. Donors with missing data included 4 HCMV+ females, 2 HCMV- females, 2 HCMV+ males and 3 HCMV- males. If all 108 infants had results for 5 cytokines for all 5 TLR ligands there would be 2,700 wells. Data were missing for 95 (3.5%) of these wells. The results tables now provide the precise number of infants tested for each condition demonstrating that the missing data came from all groups. We have also discussed the missing data further in the Statistical analysis section of Methods.

  1. While I am picking on subsets, the CMV- subsets are rather small and this limits the power of the study. Acknowledgement of this is lacking.

    Response: We have now acknowledged this limitation in the discussion.

  1. Because urine DNA were used to define CMV positivity, and not IgG, the identification of truly CMV+ patients is muddled. Urine CMV DNA suggests to this reviewer that the virus could be active in those children causing shedding, and not necessarily latent. While I understand that IgG isn’t ideal for these infants either, the strategy employed allows inclusion of CMV+ children that are not actively shedding in the CMV-negative group. Thus I am not completely convinced that all CMV+ patients are in the CMV+ group, and conversely that not all infants in the CMV- group are actually negative. This leaves the comparison as CMV-negative (includes true negative and CMV-latent not actively shedding) to CMV+ (actively infected shedding). While one might first be excited that despite inclusion of CMV+ children (latent not shedding) in the CMV-negative group there were still detectable differences, a different interpretation (detailed below) is that after active infection CMV+ children responses to TLR revert back to those of CMV-negative children.

    Response: We have data from two previous studies conducted in The Gambia by one of our co-authors (H Whittle) in which children were followed up for HCMV infection from birth. One study followed 184 babies and collected urine within 3 weeks of birth and then at three monthly intervals until 9-15 months of age (Bello and Whittle, J Clin Pathol 1991;44:366-69). They found that viral shedding was often persistent from when it is first detected, including those who were positive within the first 3 weeks of life. A study conducted in Sukuta (the same study area as the present study) tested urine for HCMV infection every 2 weeks to 3 months and then monthly to one year (Miles et al, J Virol 2007; 81(11):5766-76). This showed that 62% of infants were infected by 3 months of age rising to 76% by 9 months and that the majority of those infected in the first 3 months still excreted HCMV in their urine at 9 months. Hence, the majority of HCMV+ infants will have been detected and only a minority of the HCMV- infants are likely to have acquired infection and ceased shedding. We have now discussed these data and acknowledged this as a potential limitation in the discussion.

  1. Finally, to piggyback on #3, if CMV shedding in the DNA is indicative of active and not latent infection, then evaluating these children at later time points should give a more accurate indication of the impact of latent CMV on immune responses to vaccines earlier in life. This obviously would require significant time and resources and while not necessary for this manuscript, should be considered for future study. Nevertheless, the conclusions drawn here need to at least acknowledge the fact that many (if not all) of the CMV+ children are actively infected, and that further study is required to confirm whether the phenotype observed here is short term or long term. In other words, will the phenotype persist during latency?

    Response: As above, this limitation has now been discussed in the manuscript. It would certainly be interesting to study the effects of HCMV infection over time, but this was beyond the scope of the present study.

Minor concerns

1. P2 Row 68 techniques should be changed to mechanisms

Done

Reviewer 2 Report

The manuscript Cox et al discusses the impact of human cytomegalovirus infection on in vitro reactivity to toll-like receptor ligand stimulation in African infants. The study reveals the sex-specific differences in males and females. The study only provides a snapshot on five cytokines and at a single conc. of TLR ligands. Overall, these results though preliminary are very interesting and warrant publication. My specific comments are:

  1. I suggest the authors find a more suitable title. It is not clear if these results will hold for all African infants and authors should avoid generalization. Further, the differences are only on TLR2 and TLR7/8 stimulation which may be key to future work. The current title indicates if it overall affects TLR ligand stimulation.
  2. “Unadjusted and adjusted p-values are 160 shown in the tables.” Please provide details on how p values were adjusted in methods.
  3. “Among the HCMV+ infants, 63.3% were males, and 71.2% of were females.” Please simplify this sentence.
  4. 3“TNF-α: IL-10 ratios followed the same pattern, being significantly lower in HKLM cultures in HCMV+ females (unadjusted p = 0.038, adjusted p = 0.021) but not males, but comparable in the HCMV+ and HCMV- groups for LPS, flagellin and CLO-75 when males and females were analyzed separately (Table 6).” The sentence is too confusing. How the different groups were analyzed?
  5. A table with all details of subjects, including age, sex, HCMV copy number, and any other information will be useful. Also in which season these samples were collected is important.
  6. It is not clear if these kids had some other infection? I am particularly curious about malaria and other common bacterial and viral pathogens. Did authors check seroreactivity against malarial antigens and parasitemia in blood?
  7. In Tables 1 and 2, looking at medium only, there is a difference in male and female groups in HCMV- group for TNF-a and IL-1b? Could this be due to some other disease or unknown factors? Please comment.
  8. Why only high conc. of different TLR ligands were chosen? Looking at differences in unstimulated groups, I suspect, titration of different TLR ligands may be more useful? Please comment and discuss accordingly.

Author Response

  1. I suggest the authors find a more suitable title. It is not clear if these results will hold for all African infants and authors should avoid generalization. Further, the differences are only on TLR2 and TLR7/8 stimulation which may be key to future work. The current title indicates if it overall affects TLR ligand stimulation.

    Response: The title has been changed to the following “Sex-differential impact of human cytomegalovirus infection on in vitro reactivity to toll-like receptor 2 and 7/8 stimulation in Gambian infants”.
  2. “Unadjusted and adjusted p-values are shown in the tables.” Please provide details on how p values were adjusted in methods.
    Further detail about adjusting for covariates is now included in statistical analysis section in Materials and Methods.

  1. “Among the HCMV+ infants, 63.3% were males, and 71.2% of were females.” Please simplify this sentence.

    Response: Done
    .

  1. TNF-α: IL-10 ratios followed the same pattern, being significantly lower in HKLM cultures in HCMV+ females (unadjusted p = 0.038, adjusted p = 0.021) but not males, but comparable in the HCMV+ and HCMV- groups for LPS, flagellin and CLO-75 when males and females were analyzed separately (Table 6).” The sentence is too confusing. How the different groups were analyzed?

    Response: The sentence has been rewritten.

  1. A table with all details of subjects, including age, sex, HCMV copy number, and any other information will be useful. Also in which season these samples were collected is important.

    Response: A new table (Table 1) is now included showing donor numbers by HCMV status and sex, although we don’t have HCMV copy number details. All samples were collected during the dry season which runs from Nov to May in The Gambia. This information is now provided in Results.

  1. It is not clear if these kids had some other infection? I am particularly curious about malaria and other common bacterial and viral pathogens. Did authors check seroreactivity against malarial antigens and parasitemia in blood?

    Response: All children were well and afebrile on the day of urine and blood collection. Having a fever or being unwell were exclusion criteria. We therefore did not test for other pathogens. We have not stated this in results.

  1. In Tables 1 and 2, looking at medium only, there is a difference in male and female groups in HCMV- group for TNF-a and IL-1b? Could this be due to some other disease or unknown factors? Please comment.

    Response: It is true that males seem to have much higher levels of TNF-α and IL-1β in unstimulated (medium) blood, however, these differences were not significant. Nevertheless, it is well known that resting ex-vivo circulating cytokine levels can vary between the sexes for many reasons including differences in circulating sex hormone levels and sex differences in the microbiota and sex differences in the expression of the multiple immune response genes expressed on the X chromosome, all affecting systemic inflammation. This has now been discussed in the paper.  

  1. Why only high conc. of different TLR ligands were chosen? Looking at differences in unstimulated groups, I suspect, titration of different TLR ligands may be more useful? Please comment and discuss accordingly.

    Response: TLR ligand titrations were performed in optimization assays in order to determine the concentration required to obtain an optimal cytokine response. Given the small volumes of blood available and the need to conduct other assays as part of the vaccine trial, we were not able to test multiple concentrations. Indeed, cytokine multiplex plates are very expensive and we did not have the resources to test multiple concentrations for each TLR ligand and cytokine, so had to settle for one concentration. Furthermore, the concentrations chosen are in line with other studies measuring responses to TLR ligands in infants. We now discuss how the concentrations were chosen in Materials and Methods and cite a previously published paper in which we used the same concentrations (Burl et al, Plos One 2011).   

Reviewer 3 Report

The manuscript analyzed the sex-differential impact of human cytomegalovirus infection (HCMV) on African infants who were receiving their routine vaccination. Authors found that HCMV suppressed reactivity to TLR2 and TLR7/8 stimulation in females but not in males. It is an interesting study to show sex-differential effects on viral infection in infants. However, data shown in Tables, which presented differences of cytokine responses by TLR antagonist, are hard to interpret. If data in tables change to a figures format, the findings of the study would be better appreciated.

Comments

In line 46, authors described TLR 3,7, and 9 roles in type I IFN responses in MCMV but they referenced three HVMV papers.

Even if the study population was published before, the population data should present in a table.

In line 129, specifics of the positive and negative controls used for diagnosis of HCMV infection need to be stated rather than “appropriate” controls.

In line 136, whole blood samples were taken, and blood was used for culture. However, detail methods of whole blood culture is missing. What were the types of the blood taken? Is it PBMC? If the PBMC were cultured, how long were the cells cultured? What was number of cells used for each cytokine analysis? Is there any variation between infants for the number of cells in the 100 uL of blood?

Expression levels of each cytokine with different treatment of TLR agonists were presented by geometric mean instead of arithmetic mean. What is the rational to use geometric mean?

In line 171, among HCMV+ infants, 63.3% were male and 71.2% were female. This statement needs to be revised because it exceeds 100%.

Authors used “medium” as control in Tables. This is a confusing term. Is this culture medium? Or unstimulated control cells? If it is unstimulated cells, it would be better to use unstimulated control rather than medium.

It would be interesting to see whether the sex-differential effects against the virus infection is observed only for the early stage or throughout the life span. More mechanistic analysis of these cytokine responses by TLR agonists treatment or even TLR expression levels itself would provide better understanding on the sex-differential effect.

Minor comments

In line 35, human herpes virus 8 needs to be changed to human herpes virus 5.

In line 52, NK cells and NKT cells need to be defined.

Lines 196, 198, 201, and page 17, CLO-75 and CLO75 were used. In page 17, both “HCMV+” and “HCMV positive” were used. “CMV” were used in the tables, “HCMV” was used in text. A consistent term is needed to be used.

Author Response

The manuscript analyzed the sex-differential impact of human cytomegalovirus infection (HCMV) on African infants who were receiving their routine vaccination. Authors found that HCMV suppressed reactivity to TLR2 and TLR7/8 stimulation in females but not in males. It is an interesting study to show sex-differential effects on viral infection in infants. However, data shown in Tables, which presented differences of cytokine responses by TLR antagonist, are hard to interpret. If data in tables change to a figures format, the findings of the study would be better appreciated.

Response: The tables cover a lot of data and allow us to show the adjusted and unadjusted results. We have however made a figure of the key significant findings to further illustrate the results, as suggested by reviewer 4 as well (Figure 1).

Comments

In line 46, authors described TLR 3,7, and 9 roles in type I IFN responses in MCMV but they referenced three HVMV papers.

Response: Reference 7 discusses both human and murine CMV and the information about the use of these 3 TLRs comes from the MCMV model. The first 2 references are HCMV studies. The sentence has now been reworded so that it is clear when talking about HCMV or MCMV.

Even if the study population was published before, the population data should present in a table.

Response: This is now provided as Table 1.

In line 129, specifics of the positive and negative controls used for diagnosis of HCMV infection need to be stated rather than “appropriate” controls.

Response: These details are now provided.

In line 136, whole blood samples were taken, and blood was used for culture. However, detail methods of whole blood culture is missing. What were the types of the blood taken? Is it PBMC? If the PBMC were cultured, how long were the cells cultured? What was number of cells used for each cytokine analysis? Is there any variation between infants for the number of cells in the 100 uL of blood?

Response: The study used a whole heparinised venous blood assay in which the blood was pipetted directly into 96-well plates as stated in Materials and Methods. This assay provides a better assessment of the true ex-vivoresponse compared to analysing responses using separated PBMC, since it doesn’t require cell manipulation or addition of media, antibiotics or any other factors. The Materials and Methods section has been altered to make this more clear.

Expression levels of each cytokine with different treatment of TLR agonists were presented by geometric mean instead of arithmetic mean. What is the rational to use geometric mean?

Response: The geometric mean is more appropriate when the numbers in the dataset are not independent of each other or if they make large fluctuations. Human cytokine data are not independent since one cytokine can influence the others and there are large variations from one individual to another. Furthermore, they data are not normally distributed and thus the it would not be appropriate to report the mean values. Therefore, it was felt more appropriate to report the geometric mean rather than the arithmetic mean.

In line 171, among HCMV+ infants, 63.3% were male and 71.2% were female. This statement needs to be revised because it exceeds 100%.

Response: The sentence has been revised.

Authors used “medium” as control in Tables. This is a confusing term. Is this culture medium? Or unstimulated control cells? If it is unstimulated cells, it would be better to use unstimulated control rather than medium.

Response: We used RPMI which was also used to dilute the TLR ligands in order to ensure the volumes were equivalent in each well. In other words, the cells were unstimulated. This has now been clarified in Materials and Methods and the tables have been changed to state “Unstimulated” rather than “Medium” 

It would be interesting to see whether the sex-differential effects against the virus infection is observed only for the early stage or throughout the life span. More mechanistic analysis of these cytokine responses by TLR agonists treatment or even TLR expression levels itself would provide better understanding on the sex-differential effect.

Response: We agree that these observations should be explored further but this is outside the scope of this study.

Minor comments

In line 35, human herpes virus 8 needs to be changed to human herpes virus 5.

Response: Done.

In line 52, NK cells and NKT cells need to be defined.

Response: Done.

Lines 196, 198, 201, and page 17, CLO-75 and CLO75 were used. In page 17, both “HCMV+” and “HCMV positive” were used. “CMV” were used in the tables, “HCMV” was used in text. A consistent term is needed to be used.

Response: This has been corrected and is now consistent throughout the manuscript and tables.

Reviewer 4 Report

In this manuscript, Cox et al., analyze the impact of sex and HCMV infection in the cytokine response of infants to toll-like receptor ligands. The results presented in this paper are very valuable to understand possible sex differences in the activation of the innate response upon vaccination of newborns. HCMV is recognized as a major health concern in newborns, but how this virus may impact the efficacy of vaccines against other deadly pathogens is less understood. This paper contributes to this gap of knowledge. The manuscript is very well written and it has potential to be published in Vaccines but the authors should consider the following changes and concerns to get this work ready for publication.

  1. Authors should include a new table indicating the absolute numbers and % of the characteristics of the subjects included in this study. I am aware some of this information is included in the tables of the different cytokine levels, but the reader would benefit from a table dedicated exclusively to present the analyzed population. Also, if possible, this new table should include information about the cases with missing data, how are these distributed among the different groups of gender and HCMV infection?

  1. In Material and Methods, the authors explain that cases with missing data correspond to samples with insufficient blood volume, clotted or contaminated, and that these were substituted by multiple imputation during the analysis. However, reading the reasons provided by the authors for missing data it appears to me that cases may be missing data completely at random (MCAR) rather than at random (MAR). In studies with data MCAR, multiple imputation may not be required and may even introduce detrimental bias. Multiple imputation may be justified in data MCAR if there is a significant % (but not too high) of cases with missing data, this is why is important that the authors provide more information about the numbers and % of cases with missing data in each group. Could the authors explain why they think their cases are missing data at random and not completely at random? Have the authors performed a Little’s MCAR test to rule out the possibility that their data are MCAR? If the implementation of multiple imputation is not completely justified (MCAR with <5% or >40% of missing data), authors should reanalyze the data including only complete cases and consider including this new analysis with the results already presented in this version of the manuscript.

  1. The tables provided by the authors are very complete and informative, but authors should consider including dot-plot graphs for their more interesting results (TNF-HKLM; IL6-CLO75; IL10-CLO75). These graphs would be very helpful to convey the reader about the intragroup variation of the data and the different responses between sexes and HCMV status.

OTHER MINOR COMMENTS

  1. Line 35. Authors present HCMV as human herpesvirus 8, this is not correct. HCMV is human herpesvirus 5.

  1. Line 171. In its current form, this line reads like 63.3% of the HCMV+ infants were males, and 71.2% were females. These 2 values do not add up to 100%. I think the authors are trying to say that of all the males 63.3% were infected and of all the females 71.2% were infected. Please rewrite this line to avoid confusions.

Author Response

  1. Authors should include a new table indicating the absolute numbers and % of the characteristics of the subjects included in this study. I am aware some of this information is included in the tables of the different cytokine levels, but the reader would benefit from a table dedicated exclusively to present the analyzed population. Also, if possible, this new table should include information about the cases with missing data, how are these distributed among the different groups of gender and HCMV infection?

    Response: This information is now included as a new Table 1. The missing data were for different TLR ligands for particular infants so it is not possible to incorporate it into this table. We have therefore discussed the nature of the missing data in the Statistical analysis section of Materials and Methods and also now show the actual number of infants tested for each cytokine / stimulant pair for HCMV+ and HCMV- males and females in the tables.

  1. In Material and Methods, the authors explain that cases with missing data correspond to samples with insufficient blood volume, clotted or contaminated, and that these were substituted by multiple imputation during the analysis. However, reading the reasons provided by the authors for missing data it appears to me that cases may be missing data completely at random (MCAR) rather than at random (MAR). In studies with data MCAR, multiple imputation may not be required and may even introduce detrimental bias. Multiple imputation may be justified in data MCAR if there is a significant % (but not too high) of cases with missing data, this is why is important that the authors provide more information about the numbers and % of cases with missing data in each group. Could the authors explain why they think their cases are missing data at random and not completely at random? Have the authors performed a Little’s MCAR test to rule out the possibility that their data are MCAR? If the implementation of multiple imputation is not completely justified (MCAR with <5% or >40% of missing data), authors should reanalyze the data including only complete cases and consider including this new analysis with the results already presented in this version of the manuscript.

    Response: Given the interrelatedness between cytokine responses in individuals whereby one cytokine can influence another, it was felt important to account for missing values for those few donors for whom we did not have a result in order to obtain a complete set of data. For example, for a particular TLR ligand, we had values for one cytokine for a particular infant but not another. For this reason, the data are considered as MAR, rather than MCAR, since there is a relationship between the missing data and the observed data. It was therefore not felt necessary to run a Little’s MCAR test. The amount of missing data was small. It was complete for all 108 infants for HKLM,107 infants for unstimulated control, 104 infants for LPS, 102 for flagellin and 100 for CLO75. Donors with missing data included 4 HCMV+ females, 2 HCMV- females, 2 HCMV+ males and 3 HCMV- males. Overall, the missing data equated to 95 wells (3.5%) out of a potential of 2,700 wells (108 donors tested for 5 cytokines with 5 conditions). We have now explained this in the Statistical analysis section of the paper and changed the results tables to show the actual number of infants tested for each cytokine / TLR ligand pair for HCMV+ and HCMV- males and females. We therefore do not think it is necessary to re-analyse the data.

  1. The tables provided by the authors are very complete and informative, but authors should consider including dot-plot graphs for their more interesting results (TNF-HKLM; IL6-CLO75; IL10-CLO75). These graphs would be very helpful to convey the reader about the intragroup variation of the data and the different responses between sexes and HCMV status.

    Response: New dot plots are now provided as suggested as a new Figure 1.

OTHER MINOR COMMENTS

  1. Line 35. Authors present HCMV as human herpesvirus 8, this is not correct. HCMV is human herpesvirus 5. 

    Response: This has been corrected.

  1. Line 171. In its current form, this line reads like 63.3% of the HCMV+ infants were males, and 71.2% were females. These 2 values do not add up to 100%. I think the authors are trying to say that of all the males 63.3% were infected and of all the females 71.2% were infected. Please rewrite this line to avoid confusions.

    Response: This sentence has been re-written.

Round 2

Reviewer 1 Report

My previous concerns have been addressed.  However, there was a question posed by one of the other reviewers regarding cell counts that was not addressed. This is an impt question that cannot be dodged. Humans have significant variability in their white blood cell counts. Because 100 ul samples of whole blood were used and cell numbers were not intentionally controlled, it is possible that differences in TLR- responsiveness observed were a consequence of lower or higher cell counts and not because of actual differences in TLR responsiveness. For example, if females had on average lower cell counts than males, one might observe a difference in TLR responsiveness simply from having fewer cells in those wells.

If white blood cell counts are available from the day of blood collection then the stimulation results should be normalized relative to WBC and re-evaluated. If not, a different way to control for this would be to look at relative responses in individuals. IE TLR-2 vs 4 vs 7 for each person, and not the means of all TLR responses. One of the responses could be “normal”, with remaining results compared relative to this normal. Alternatively they could be normalized to unstimulated cells, which is probably a better “normal”.

Without such analysis, it is unclear whether the differences are real, or simply artifact of variable cell counts between subjects.

Minor

In figure 1, for clarity sake I would suggest labeling each graph with the TLR- (ie TLR-4 etc) instead of the ligand used. Would help the tables too. This would save the reader the mental math of remembering which ligand goes with which receptor.

Author Response

My previous concerns have been addressed.  However, there was a question posed by one of the other reviewers regarding cell counts that was not addressed. This is an impt question that cannot be dodged. Humans have significant variability in their white blood cell counts. Because 100 ul samples of whole blood were used and cell numbers were not intentionally controlled, it is possible that differences in TLR- responsiveness observed were a consequence of lower or higher cell counts and not because of actual differences in TLR responsiveness. For example, if females had on average lower cell counts than males, one might observe a difference in TLR responsiveness simply from having fewer cells in those wells. If white blood cell counts are available from the day of blood collection then the stimulation results should be normalized relative to WBC and re-evaluated.

Response: The whole-blood assay (WBA) is a widely accepted immunological assay for measuring human cytokine responses to antigenic stimulation. The assay is considered more physiological and representative of true in vivo reactivity than when measuring responses using separated PBMC (Silva et al, 2013). This is because the cells are present in their natural frequencies and proportions in vivo. The WBA can be used to measure T cell memory recall responses or to measure innate immunity. T cell assays generally require several days of stimulation in order to detect optimal responses, while the innate assays are generally conducted for shorter periods of <24 hours, as used in our study. The WBA has been validated for detecting innate responses in short-term cultures in previous studies (Punsmann et al, 2013; Liebers et al, 2018) and we have published many studies using whole blood assays using shorter incubation periods to detect innate reactivity to TLR ligand stimulation (Burl et al, 2010; Jensen et al, 2015; Noho-Konteh et al, 2016; Freyne et al, 2028; Freyne et al, 2020) and longer incubation periods of 3-5 days for detecting T cell responses by WBA (Burl et al, 2011).

The WBA is considered a measure of the capacity of a defined volume of blood to mount an immune response, and we do not know the relative contribution of cytokine from each of the constituent cell types. The cells contributing to the cytokine response in the innate WBA includes activated monocytes, macrophages, dendritic cells, NK cells, NK T cell, γδ T cells and neutrophils. Generally, memory B and T cells will not be involved. The white cell count is a composite measure of lymphocytes and neutrophils with a small contribution from monocytes, eosinophils and basophils; therefore, adjusting for WBC may not be helpful given the relative contributions of different WBC subsets to the innate response. For this reason, immunologists do not count and control for the number of white cells or lymphocytes in the WBA. By contrast, assays involving isolated peripheral blood mononuclear cells (PBMCs) need to counted to use an equivalent number of PBMC resuspended in medium per well to allow for consistency. The WBA captures the global response to the immune stimulus, whereas PBMC assays measure the ability of a set number of isolated PBMC (lymphocytes and monocyte/macrophages) to respond to a stimulus, which is less physiological. That said, several studies have shown a good correlation between WBA and PBMC stimulation assays, but mainly for T cell memory assays (Deenadayalan et al, 2013; Silva et al, 2013; Mohammadi et al 2019). For these reasons we do not think that normalising for the WBC is required.

We have now discussed some of these aspects of the WBA in the limitations paragraph in the discussion.

References

Burl S, M Cox, J Adetifa, E Touray, MO Ota, A Marchant, H McShane, H Whittle, S Rowland-Jones, KL Flanagan. Delaying BCG vaccination from birth to 4½ months of age reduces the induction of potentially protective mycobacterial specific Th1 and IL-17 responses. J Immunol 2010,185(4): 2620-2628.

Burl S, J Townend, J Njie-Jobe, M Cox, U J Adetifa, E Touray, A Jaye, V Philbin, C Mancuso, B Kampmann, H Whittle, KL Flanagan*, O Levy*. Age dependent maturation of Toll-Like Receptor-mediated cytokine responses in Gambian infants. PLoS One 2011; 6(4): e18185. *Joint last authors

Deenadayalan A et al. Comparison of whole blood and PBMC assays for T-cell functional analysis. BMC Research Notes 2013; 6, 120. 

Freyne B, S Donath, S Germano, K Gardiner, D Casalaz, RM Robins-Browne, N Amenyogbe, N Messina N, MG Netea, KL Flanagan, T Kollmann, N Curtis. Neonatal BCG Vaccination Influences Cytokine Responses to Toll-like Receptor Ligands and Heterologous Antigens. J Infect Dis 2018; 217(11): 1798-1808.

Freyne B, N Messina, S Donath, S Germano, R Bonnici, K Gardiner, D Casalaz, RM Robins-Browne, MG Netea, KL Flanagan, T Kollmann, N Curtis. Neonatal BCG vaccination reduces interferon-y responsiveness to heterologous pathogens in infants from a randomized controlled trial. J Infect Dis 2020; 221(12): 1999-2009.

Jensen KJ, N Larsen, S Biering-Sørensen, A Andersen, HB Eriksen, I Monteiro, D Hougaard, P Aaby, MG Netea, KL. Flanagan, CS Benn. Heterologous immunological effects of early BCG vaccination in low-birth weight infants in Guinea-Bissau: a randomized-controlled trial. J Infect Dis 2015; 211(6): 956-67.

Liebers V et al. Cell Activation and Cytokine Release Ex Vivo: Estimation of Reproducibility of the Whole-Blood Assay with Fresh Human Blood. Adv Exp Med Biol 2018; 1108: 25-36.

Mohammadi A et al. Comparison of cytokine profile of IFN-γ, IL-5 and IL-10 in cutaneous leishmaniasis using PBMC vs. whole blood. Iran J Microbiol 2019; 11(5): 431-39.

Noho-Konteh F, JU Adetifa, M Cox, S Hossin, J Reynolds, MT Le, LC Sanyang, A Drammeh, T Forster, P Dickinson, P Ghazal, M Plebanski, H Whittle, SL Rowland-Jones, JS Sutherland, KL Flanagan. Sex-differential non-vaccine specific immunological effects of diphtheria-tetanus-pertussis and measles vaccination. Clin Infect Dis 2016; 63(9): 1213-26.

Punsmann et al. Ex vivo cytokine release and pattern recognition receptor expression of subjects exposed to dampness: pilot study to assess the outcome of mould exposure to the innate immune system. PLoS One 2013; 8(12):e82734.

Silva et al. A whole blood assay as a simple, broad assessment of cytokines and chemokines to evaluate human immune responses to Mycobacterium tuberculosis antigens. Acta Tropica 2013; 127(2): 75-81.

If white blood cell counts are available from the day of blood collection then the stimulation results should be normalized relative to WBC and re-evaluated. If not, a different way to control for this would be to look at relative responses in individuals. IE TLR-2 vs 4 vs 7 for each person, and not the means of all TLR responses. One of the responses could be “normal”, with remaining results compared relative to this normal. Alternatively they could be normalized to unstimulated cells, which is probably a better “normal”.

Without such analysis, it is unclear whether the differences are real, or simply artifact of variable cell counts between subjects.

Response: We believe that our analysis approach has done exactly what the reviewer is suggesting. In the adjusted regression analyses, each TLR-agonist geometric mean difference estimate was adjusted for each of the other TLR-agonists. Z-scores of those other TLR-agonists were used for the adjustment process: z-score is (subject value minus mean value) divided by standard deviation. In regression analysis, the constant value (i.e. the mean for example of the females without CMV) is adjusted back to the value where all other variables in the model are zero. The effects of gender and CMV are estimated from the regression coefficients produced by the regression model. Thus, consistent with your suggestion, the individual scores of each infant are adjusted to the other TLR-agonist values to allow for potential imbalances in those other values. Since we used natural logarithmic transformed values to estimate the geometric means, the relationship between the outcome TLR-agonist responses and the predictor adjustment TLR-agonist responses in the regression model is that of individual ratios for each of the infant participants.

We have further clarified this in the Statistical analysis section of the paper.

Minor

In figure 1, for clarity sake I would suggest labeling each graph with the TLR- (ie TLR-4 etc) instead of the ligand used. Would help the tables too. This would save the reader the mental math of remembering which ligand goes with which receptor.

Response: This has been changed as suggested.

All new changes are highlighted in turquoise.